# Meta Compression: Learning to Compress Pre-trained Deep Neural Networks

## Abstract

State-of-the-art deep neural networks (DNN) have achieved outstanding results in a variety of tasks. Unfortunately, these DNN are so large that they cannot fit into the limited resources of edge servers or end devices such as smartphones and IoT sensors. Several approaches have been proposed to design compact yet efficient DNNs, however, the performance of the compressed model can be only characterized a posteriori. This work addresses this issue by introducing meta compression, a novel approach based on meta learning to simplify a pre-trained DNN into one that fulfills given constraints on size or accuracy. We also leverage diffusion-based generative models to improve generalization performance of meta learning and extensively evaluate meta compression on an image classification task with popular pre-trained DNNs. The obtained results show that meta compression achieves a 92% top-5 recommendation accuracy and that the top-1 recommendation is only 1% far from the optimal compression method in terms of average accuracy loss.

## 1 Introduction

Deep neural networks (DNNs) are being increasingly adopted in diverse domains including computer vision (Redmon et al., 2016), natural language processing (Vaswani et al., 2017), and speech recognition (Amodei et al., 2015). The size of state-of-the-art DNNs has exponentially grown in recent years (Desislavov et al., 2023), allowing them to achieve or even exceed human-level performance for a variety of tasks (Alzubaidi et al., 2021; Silver et al., 2018). Despite this unquestionable benefit, these DNNs have become so large that in some cases they cannot run on a single GPU. Smaller models cannot still fit the limited resources of edge servers or end devices such as mobile phones and smart objects in the Internet of things. This work specifically addresses such a challenge to enable efficient computation, closer to end users.

Several approaches have been proposed to design compact yet efficient DNNs – in terms of both accuracy and performance – by explicitly targeting resource-constrained devices, for instance. On the one hand, model compression takes an existing DNN and applies techniques such as pruning and (or) quantization to obtain a simplified model (Gale et al., 2019; Gholami et al., 2022). The obtained model is smaller, therefore, also faster to execute for inference. This approach has been shown to obtain a significant reduction in model size with close to negligible loss in accuracy (Han et al., 2015). Despite its potential, the performance of the compressed model can be only characterized a posteriori: a source DNN has to be first compressed and only then evaluated. As a consequence, tailoring a model to fulfill certain requirements results in a trial-and-error process that depends on appropriately setting the parameters of the specific compression method such as target sparsity level, quantization precision, fine-tuning schedule, and so on.

On the other hand, neural architecture search explores a design space to find a model that satisfies application-specific requirements in an automated way (Chitty-Venkata & Somani, 2022). This approach effectively enables tailoring the characteristics of a DNN architecture, for instance, based on hardware-specific constraints. However, it entails carrying out an extremely large amount of computation to explore a combinatorially large design space (Wu et al., 2019; Stamoulis et al., 2019). Even worse, the process generally needs to be repeated from scratch for each target configuration. Recently, more advanced techniques have been introduced to train a so-called once-for-all network (Cai et al., 2019) that can be specialized for diverse

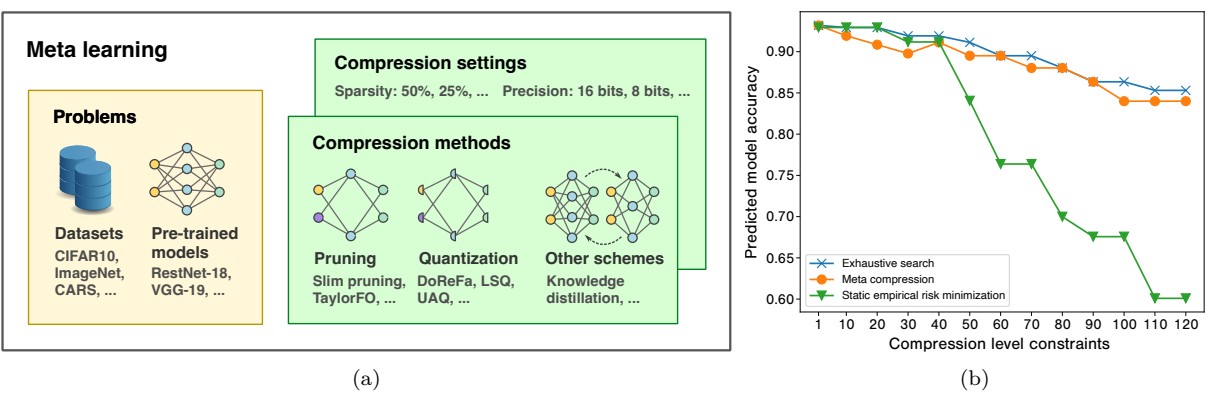

(a)            (b)

Figure 1: (a)Given several datasets and models pre-trained on them, meta compression predicts the accuracy of a source model when compressed with different methods and settings, *without* running it. (b) Meta compression outperforms a recommendation strategy based on empirical risk minimization for high compression levels (at least 50 times), with a close-to-optimal prediction accuracy.

target configurations without re-training. However, these schemes significantly constrain the design space and still require substantial computation (Section 2).

This article bridges the gap between these two extremes by introducing **meta compression**, a novel approach based on meta learning to simplify a source DNN into one that fulfills application- and device-specific constraints. Our approach aims at answering the following questions: Which is the simplest DNN that achieves a certain accuracy? Which is the most accurate DNN that can run on a given device? It does so by predicting the compressed model accuracy of given pre-trained DNN models. This is achieved by learning across a collection of problems, consisting of several reference DNN models, compression methods, and their settings (Figure 1).

Employing pre-trained DNN models is a key design choice to reduce the computational overhead of meta learning. However, such a choice entails several challenges related to the partitioning of the source dataset the DNN was trained on. In fact, meta learning requires access to sufficient evaluation data to estimate generalization performance of compressed models. Evaluation data should also be separate from the test data used to evaluate the performance of the meta learning. In a practical deployment scenario, the training / testing split may not be known for a given pre-trained model, or the data not used for training may be not enough for meta learning. For this reason, we leverage diffusion models when needed to improve the generalization performance of meta learning and also achieve a high prediction accuracy. We extensively evaluate our proposed framework by using popular DNNs, achieving a 92% top-5 recommendation accuracy compared to the 15% that is obtained with the compression algorithm that works the best on average. Moreover, our top recommendation is only 1% far from the optimal compression method in terms of average accuracy loss.

In summary, our work establishes the following contributions.

- We introduce a meta compression framework that takes a collection of pre-trained models and a set of compression methods so as to predict the accuracy of a compressed model without the need for evaluating it. Our framework obtains the most accurate option that fulfills given resource constraints. We also provide theoretical guarantees on the effectiveness of meta learning across multiple problems and compression methods, in terms of both scalability and maximum prediction error (Section 3).

- We extensively evaluate meta compression on an image classification task with models pre-trained on the CIFAR10 and ImageNet datasets. The obtained results show that the learned meta predictor is indeed accurate: it reliably estimates the performance of new compressed architectures after compression by leveraging features specific to the DNNs and to the compression methods, in addition to evaluation data generated by diffusion models (Section 4).

## 2 Related Work

**Model compression**. DNN compression broadly encompasses techniques aiming at simplifying a source model into a smaller one (Hoefler et al., 2021). Among them, the major approaches are quantization, to reduce the precision of operands, and pruning, to sparsify a network by removing "unimportant" weights or activation values (Gholami et al., 2021; Liang et al., 2021). Both benefit from fine-tuning to restore accuracy (Sanh et al., 2020) and can jointly be applied for further reduction in model size (Hu et al., 2021; Yang et al., 2020; Frantar & Alistarh, 2022). However, characterizing the performance of specific compression methods have been elusive. Kuzmin et al. (2023) introduce an analytical framework to compare pruning and quantization. However, their results are limited to magnitude pruning and uniform quantization as considered separately. In contrast, our solution allows to predict the accuracy of diverse compression methods, including those leveraging fine-tuning, even when jointly applied.

**Network architecture search**. Network architecture search aims at automatically selecting a model that can satisfy certain properties (Elsken et al., 2019). In this broad context, AutoNBA (Fu et al., 2021) devises a search strategy over networks, bitwidths, and features for hardware acceleration. To reduce computation, OFA (Cai et al., 2019) takes a reference DNN as a backbone to build a so-called once-for-all network. Such a network is trained in such a way that it can be tailored for hardware-specific constraints at a later stage, without re-training – the latter phase achieves pruning as a side effect. The same approach is extended in APQ (Wang et al., 2020) to also consider quantization. However, APQ only supports channel pruning and uniform quantization. Instead, our approach allows to plug-in arbitrary pruning and quantization schemes as compression methods.

**Meta learning**. Meta learning involves iterating over ML tasks while an outer training loop optimizes meta parameters (Hospedales et al., 2022) and has been applied to different scenarios (Garg & Kalai, 2018; Verma et al., 2020). Metapruning (Liu et al., 2019) applies meta learning to predict weights of a channel-pruned DNN, which is in turn employed to derive a specific channel pruning strategy. Instead, our approach applies meta learning to predict the performance of a compressed model and is not limited to pruning channels. In a different context, a few solutions have recently leveraged diffusion models in the context of meta learning. Among them, Meta-DM (Hu et al., 2023) performs data augmentation to improve the performance of few-shot learning, whereas the work in (Nava et al., 2023) carries out zero-shot task adaptation with both classifier and classifier-free guidance. In contrast, we introduce a framework for meta compression where diffusion models help generate evaluation data to improve prediction performance.

## 3 Meta Compression

We now introduce the system model and describe our meta compression scheme (Figure 1). For clarity, we first address a single dataset and compression method, then extend our analysis to different configurations.

### 3.1 Overview and system model

A source *dataset* $D = \{T, F, E, V\}$ consists of different, disjoint sets: $T$ for training, $F$ for fine-tuning, $E$ for evaluation, and $V$ for testing. For simplicity, we focus on classification tasks, even though our formulation is general enough to cover other domains (see Section 5 for more details). Accordingly, a *classifier c* is pre-trained on $T$ and a *problem* is defined as $p = (c, F, E)$. A *compression method K* uses the pre-trained classifier $c$ and the fine-tuning dataset $F$ to obtain a compressed classifier $c' = K(c, F)$. The loss of the compressed classifier is assessed over the evaluation set $E$ to obtain the true accuracy $A$. An *accuracy prediction model* (meta predictor) $g$ takes meta features (i.e., those derived by the meta compression) as input to derive the predicted accuracy $\tilde{A}$.

Accordingly, the training of the accuracy prediction model follows the process in Figure 2a. Meta feature extraction involves computing problem features, compression method features, and compressed classifier performance. A given classifier $c$ compressed by using a configuration $i$ with $K_i$, then the performance of the compressed classifier $c'_i$ is assessed by using the evaluation dataset $E$. In summary, training $g$ involves learn-

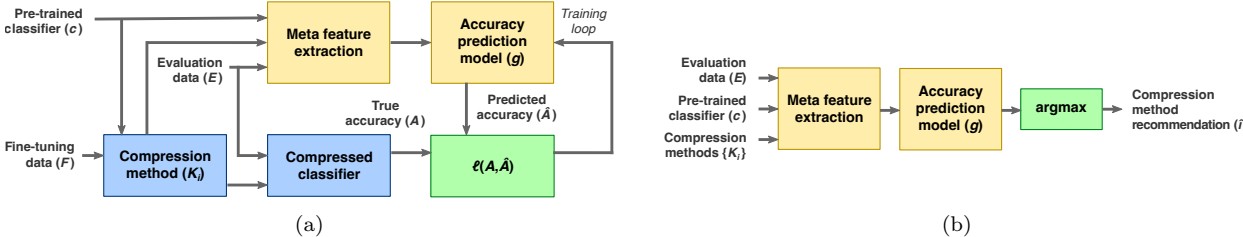

(a)                                              (b)

Figure 2: (a) The inner loop of the meta learning process involves training a model to predict the performance of a classifier once compressed with a certain method and related settings. (b) Once learned, the accuracy prediction model can be leveraged to recommend the best compression method for a given problem.

ing a mapping from problem-specific and compression method-specific features to the compressed classifier performance.

We then proceed to the recommendation once the accuracy prediction model is trained. The first type of recommendation involves selecting the most accurate compression model that satisfies device-specific constraints. To do so, we first prepare a shortlist of compression methods along with their settings. We then refine the candidate options by only keeping those that satisfy the given constraints, for instance, in terms of model size or target compression ratio. The best compression method can be selected by a simple $\arg\max$ function over the predicted accuracy values (Figure 2b).

The second type of recommendation entails selecting the method that obtains the highest compression while satisfying an application-specific accuracy target. For this purpose, we consider all available compression methods as predictions and prepare a shortlist of those fulfilling the accuracy constraint. Similar to the previous case, we use the $\arg\max$ function to obtain the method that achieves the highest predicted compression.

We have just outlined of how meta compression allows to answer the research questions introduced in Section 1. The next step is to establish a rigorous theoretical foundation by leveraging the Probably Approximately Correct (PAC) learning framework to obtain theoretical guarantees for making meta learning effective across multiple problems and compression methods.

## 3.2 Theoretical foundation

So far, we have limited our attention to a single problem $p$ and a given compression method $K$, specialized into $K_i$ according to different configurations. We now extend our consideration to a collection of problems and compression methods, and precisely state our objective. The purpose of the discussion provided next is to analytically characterize the scalability of the learning (Lemma 3.1) and the maximum prediction error achieved by meta compression irrespective of the actual input data distribution (Theorem 3.2).

In classification problems, $D$ is a labeled dataset $(\mathcal{X}, \mathcal{Y})$ and $c = L(T)$ is a classifier obtained by learner $L : (\mathcal{X}, \mathcal{Y})^* \to \mathcal{Y}^{\mathcal{X}}$ (namely, the set of all functions from $\mathcal{X}$ to $\mathcal{Y}$) based on the training set $T \sim \mu^n$. A function $l : \mathcal{Y} \times \mathcal{Y} \to [0, 1]$ is such that $l(y, \hat{y})$ is the loss when the prediction is $\hat{y}$ and the true label is $y$. $l_\mu(c) = \mathbb{E}_{(x,y)\sim\mu} [l(y, c(x)]$ characterizes the expected loss of a classifier over some data distribution $\mu$.

Now, consider a compression method $K : \mathcal{Y}^{\mathcal{X}} \times (\mathcal{X} \times \mathcal{Y})^* \to \mathcal{Y}^{\mathcal{X}}$ such that $c' = K(c, F)$, wherein the index $i$ is dropped from $K_i$ for convenience. The loss of the compressed classifier can be evaluated by using the evaluation dataset $E$. Let us consider a distribution over problems $p \sim \nu$ to evaluate the loss of a compression method $K$. Each problem specifies a pre-trained classifier $c \in \mathcal{Y}^{\mathcal{X}}$, and retraining and evaluation dataset $F, E \in (\mathcal{X} \times \mathcal{Y})^*$. Then, the expected loss of a compression method can be defined as:

$$l_\nu(K) = \mathbb{E}_{p\sim\nu} [l(y, K(c, F)(x))]$$

Let us consider the task of selecting the best performing compression method across several problems, given a family of compression methods $\mathcal{K}$ that satisfy certain constraints on the compression level. A practical objective is to select the compression method with the lowest empirical error across multiple problems,

namely, to carry out an empirical risk minimization (ERM). Accordingly, let us consider a collection $P$ of problems, containing problems $(c_\tau, F_\tau, V_\tau) = p_\tau \in P$. Then,

$$\text{ERM}_\mathcal{K}(P) \in \arg\min_{K \in \mathcal{K}} \sum_{p_\tau \in P} \sum_{(x_t, y_t) \in E_\tau} l(y_t, K(c_\tau, F_\tau)(x_t))$$

**Lemma 3.1.** *For any finite family $\mathcal{K}$ of compression algorithms, any distribution $\nu$ over problems $p_\tau$, and any $n \geq 1$, $\delta > 0$,*

$$\Pr_{P \sim \nu^n} \left[ l_\nu \left( \text{ERM}_\mathcal{K}(P) \right) \leq \min_{K \in \mathcal{K}} l_\nu(K) + \sqrt{\frac{2}{n} \log \frac{|\mathcal{K}|}{\delta}} \right] \geq 1 - \delta \tag{1}$$

*Proof.* Let

$$\epsilon = \sqrt{\frac{2}{n} \log \frac{|\mathcal{K}|}{\delta}}.$$

Consider an empirical estimate of $l_\nu(K)$ using some dataset of problems $P \sim \nu^n$, $(c_\tau, F, E) = p_\tau \in P$, given by

$$l_P(K) = \frac{1}{n} \sum_{p_\tau \in P} \sum_{(x_t, y_t) \in E} l(y_t, K(c_\tau, F_\tau)(x_t)).$$

Denote the optimal compression method $K_0 \in \arg\min_{K \in \mathcal{K}} l_\nu(K)$. Writing the Chernoff bound for $l_P(K_0)$, we have

$$\Pr_{P \sim \nu^n} [l_P(K_0) \geq l_\nu(K_0) + \epsilon/2] \leq e^{-2n(\epsilon/2)^2}.$$

Now, consider the set $B$ of bad compression methods that are more than $\epsilon$ far away from $l_\nu(K_0)$. More formally, $B = \{K \in \mathcal{K} \mid l_\nu(K) \geq l_\nu(K_0) + \epsilon\}$. By writing the Chernoff bound for each $K \in B$, we have

$$\Pr_{P \sim \nu^n} [l_P(K) \leq l_\nu(K) - \epsilon/2] \leq e^{-2n(\epsilon/2)^2}.$$

Clearly, $l_\nu(\text{ERM}_\mathcal{K}) \geq l_\nu(K_0) + \epsilon$ holds only if $l_D(K) \leq l_D(K_0)$ for some $K \in B$. Specifically, either $l_P(K) \leq l_\nu(K) - \epsilon/2$ for some $K \in B$, or $l_D(K_0) \geq l_\nu(K_0) + \epsilon/2$. By applying the union bound, the result holds with probability at most $|\mathcal{K}| e^{-2n(\epsilon/2)^2} = \delta$. $\square$

The lemma implies that the best compression method can be found across *all problems* by using $\mathcal{O}(log(|\mathcal{K}|))$ training problems. It is also possible to select the compression method that is best suited to a specific problem. Hence, let us consider the task of choosing the best compression method among candidates $\{K_1, \ldots, K_m\}$ given the problem specification $p_\tau$. For this purpose, let us consider problem features $\phi(c_\tau, F_\tau, E_\tau) \in \Phi$ and compression method features $\gamma(K) \in \Gamma$. Moreover, let us consider a family $\mathcal{F}$ of functions $f : \Phi \times \Gamma^m \to \{1, \ldots, m\}$ that apply compression method among $m$ based on extracted features. Finally, the objective of finding the function that selects the best compression method based on ERM can be defined as:

$$\arg\min_{f \in \mathcal{F}} \sum_{p_\tau \in P} \sum_{(x_t, y_t) \in E_\tau} l(y_t, K_{f(\phi(p_\tau), \gamma(K_1), \ldots, \gamma(K_m))}(c_\tau, F_\tau)(x_t)) \tag{2}$$

Solving $\text{ERM}_\mathcal{F}$ entails a multi class classification problem based on problem features $\phi(c_\tau, F_\tau, E_\tau)$ and compression method features $\gamma(K)$, while $\text{ERM}_\mathcal{K}$ selects the best single class which we call the *static ERM strategy*.

**Theorem 3.2.** *The class $\{K_i | i \in [1, m]\}$ of $m$ compression methods can be agnostically learned. In particular, a polynomial-time algorithm achieves an error of at most*

$$\min l_\nu(K_i) + \sqrt{2/n(\log m/\delta)}$$

*with probability $\geq 1 - \delta$ over $n$ training problems.*

*Proof.* Lemma 3.1 implies that, given $n$ training problems and $|K_i| \leq m$, the error of ERM is within $\sqrt{2/n \log m/\delta}$ of $\min_i l_\nu(K_i)$. Let us compute the mean loss of $K_i$ across the training problems, for each compression method $i \in [1, m]$, to find the best compression method in polynomial time (of the input size $|T|$). As $K_i$ runs in polynomial time, the loss can be computed in polynomial time too, and the number of compression methods is also bounded. Thus, the entire procedure takes polynomial time. $\qquad\square$

The theorem establishes the learnability of the compression method recommendation function $f$. We further simplify the design of $f$ by employing a loss prediction function $g : \Phi \times \Gamma \to \mathbb{R}_{\geq 0}$ that predicts the loss of a compressed classifier $c'$ for a given problem $p_\tau$ and compression method $K_i$:

$$g(\phi(p_\tau), \gamma(K_i)) \approx l_\mu(K_i(c_\tau, F_\tau))$$

Accordingly, the ERM objective for learning $g$ is

$$\underset{g \in \mathcal{G}}{\arg\min} \sum_{p_\tau \in P} \sum_i \left[ g(\phi(p_\tau), \gamma(K_i)) - l_{\mu^e}(K_i(c_\tau, F_\tau)) \right]^2 \tag{3}$$

Once $g$ has been learned by using sufficient data, $f$ can be described in terms of $g$ as:

$$f(\phi(p_\tau), \gamma(K_1), \ldots, \gamma(K_m)) = \underset{i=\{1,\ldots,m\}}{\arg\min} \ g(\phi(p_\tau), \gamma(K_i)) \tag{4}$$

This approach finds the best compression method for any problem $d_\tau$. Consequently, it also finds the best compression methods across all problems, which was the ERM objective of learning $f$. Such a formulation of $f$ has an additional benefit, as it allows us to solve this performance-constrained compression maximization problem too with the same loss prediction function $g$. We choose classification accuracy as the negative of loss to be predicted using $g$, and refer to $g$ as the accuracy predictor in the rest of the paper.

### 3.3 Meta features and accuracy predictor

We employ gradient boosted decision trees (Chen & Guestrin, 2016) as the accuracy prediction model according to (White et al., 2021), which showed that they are remarkable in predicting the performance DNNs for NAS without substantial computation. We have also empirically confirmed that employing a feed-forward DNN did not improve the results. Appendix A details the configuration of the boosted decision tree used in our evaluation, along with the considered features and their importance.

Obtaining a compressed classifier requires applying a compression method $K$ to a pre-trained classifier $c$ and possibly retraining using tuning dataset $F$. Thus, making predictions about the compressed classifier performance requires problem features $\phi(p_\tau)$ and compression method features $\gamma(K_i)$. We also need an evaluation dataset $E$ that is separate from the tuning dataset to accurately evaluate the behavior of the compressed classifier. We overview these features next.

**Problem features** $(\phi(p_\tau) \in \Phi)$. A problem $p_\tau = (c_\tau, F_\tau, E_\tau)$ consists of a pre-trained classifier, and tuning dataset, and an evaluation dataset. We extract two set of features to describe the pre-trained classifier: architecture features and solution features. Architecture features encode information about the modules used as building blocks of the DNN architecture. Instead, solution features describe the particular solution learned from training the model. These include norms of weights, gradients, in addition to loss and accuracy of $c$ evaluated through $E$.

**Compression method features** $(\gamma(K_i) \in \Gamma)$. The compression process typically consists of iterative pruning and retraining, followed by quantization and retraining. We consider several popular pruning and quantization methods in our experiments. Each pruning method is encoded via a pruning identifier and the target sparsity level. Similarly, each quantization method is encoded with a unique quantization identifier and the target level of bits.

**Compressed classifier performance**. We evaluate the loss and accuracy of the compressed classifier on the evaluation dataset $E$ considered as the ground truth for predictions.

The accuracy prediction model $g$ takes problem features and compression method features as input to predict the compressed classifier performance. After training, the predictor is able to estimate accuracy for arbitrary pruning and quantization levels. Based on this, the configurations with the highest predicted accuracy are recommended for actual compression.

# 4 Experimental evaluation

We conduct extensive experiments to evaluate the effectiveness of the proposed meta compression framework in estimating the performance of common pruning and quantization methods. The evaluation focuses on image classification tasks for models pre-trained on two datasets, namely, CIFAR10 (Krizhevsky, 2009) and ImageNet (Deng et al., 2009). We also characterize the impact evaluation data selection and the generalization performance of the meta predictor to new evaluation data, new architectures, and new compression methods. Finally, we provide results for the training costs. The code to reproduce the experiments is available at `https://anonymous.4open.science/r/DeeperCompression-4EED/`.

## 4.1 Experimental setup

### 4.1.1 Datasets and architectures

Our setup include popular DNN models trained on the CIFAR10 dataset, namely, VGG19 (Simonyan & Zisserman, 2014), ResNet18 (He et al., 2015b), GoogLeNet (Szegedy et al., 2014), DenseNet121 (Huang et al., 2016), ResNeXt29-2x64d (Xie et al., 2016), MobileNet (Howard et al., 2017), MobileNetV2 (Sandler et al., 2018), DPN92 (Chen et al., 2017), SENet18 (Hu et al., 2017), ShuffleNetV2 (Ma et al., 2018), RegNetX-200MF (Radosavovic et al., 2020), SimpleDLA (Yu et al., 2017), through their implementation in (Kuang-Liu, 2017). Unless stated otherwise, $F$ consists of 20% randomly sampled images from the CIFAR10 training dataset, $E$ consists of 10k images generated using a diffusion model trained on CIFAR10 train dataset, and $V$ comprises the complete CIFAR10 test dataset.

In addition, we considered the following architectures pre-trained on the ImageNet dataset Deng et al. (2009): VGG11 (Simonyan & Zisserman, 2015), Squeezenet (Iandola et al., 2016), Densenet121 (Huang et al., 2016), Alexnet (Krizhevsky et al., 2012), Resnet (He et al., 2015a) and Shufflenet (Zhang et al., 2017). Here $F$ consists of 30k labelled images taken from the ImageNet training set, $V$ (30k images) and $E$ (20k images) comprise images sampled from the ImageNet validation set (50k images) in the ratio of 3:2, respectively. Different from CIFAR10, we did not employ diffusion models as the source dataset contained enough samples.

### 4.1.2 Compression methods

We selected a broad set of compression methods that span complementary design paradigms and are applicable across diverse DNN architectures. A large number of pruning and quantization techniques exist, however, integrating them – especially together with retraining – poses engineering challenges because the individual stages can be mutually incompatible (Qu et al., 2025). For this reason, we evaluated a large number of options and eventually selected the methods discussed next, as they were able to consistently and reliably compress the pre-trained models considered in our study.

**Pruning**. We consider approaches entailing both unstructured and structured pruning. Specifically, we selected the following unstructured pruning methods: *magnitude pruning*, consisting of a simple weight pruning scheme; and *pruning+tuning*, as the former followed by retraining for 40 epochs. In addition, we consider the following structured pruning techniques: *L1 norm pruning* (Li et al., 2016), which removes weights with low L1 norms; *network slimming* (Liu et al., 2017), which masks scaling factors in later batch normalization layers to prune channels in convolution layers; and *TaylorFO* (Molchanov et al., 2019), which prunes convolutional layers based on the first-order Taylor expansion on the weights.

**Quantization**. We considered the following quantization methods: *uniform affine quantizer* (Krishnamoorthi, 2018), which maps continuous values to discrete levels by scaling and rounding within a given quantization range; *Dorefa-net* (Zhou et al., 2016), which represents the weights and activations through a ternary code to save storage, then employs a mix of fixed-point quantization, scaling, and rounding to reduce quantization

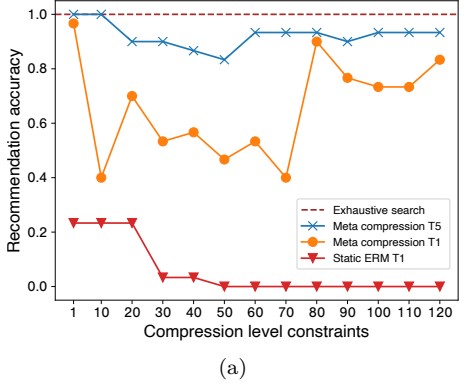

|            | (a) |            |            |            | (b) |

|            |     |

| Metric | Meta Compression | Static ERM |
|---|---|---|
| T5 Accuracy | **0.92** | 0.15 |
| T1 Accuracy | **0.66** | 0.06 |
| T1 Error | **0.01** | 0.25 |
| MAE | **0.10** | 0.12 |
| Kendall's Tau ($\tau$) | **0.65** | – |

Figure 3: (a) Recommendation accuracy of meta compression against the optimal static ERM and exhaustive search for different compression levels. (b) Recommendation performance summary metrics comparing meta compression with the static ERM approach (the single best pruning and quantization methods used across all constraint levels).

| Compression Algorithms | T5 Accuracy | T1 Accuracy | T1 Error | MAE |
|---|---|---|---|---|
| Prune + Quant | 0.97 | 0.92 | 0.02 | 0.14 |
| Level-prune + Dorefa-quant | 0.97 | 0.65 | 0.04 | 0.07 |
| L1-prune + Dorefa-quant | 0.93 | 0.45 | 0.04 | 0.08 |
| TaylorFO-prune + Dorefa-quant | 0.85 | 0.38 | 0.06 | 0.08 |
| Slim-prune + Dorefa-quant | 0.99 | 0.70 | 0.03 | 0.18 |
| Level-prune + LSQ-quant | 0.98 | 0.78 | 0.01 | 0.02 |
| L1-prune + LSQ-quant | 0.99 | 0.64 | 0.02 | 0.06 |
| TaylorFO-prune + LSQ-quant | 1.00 | 0.64 | 0.03 | 0.07 |
| Slim-prune + LSQ-quant | 0.98 | 0.72 | 0.02 | 0.17 |

Table 1: MAE of the predictor $g$ trained for different compression algorithms.

error; *learned step-size quantization* (Esser et al., 2020), which employs a learnable scaling factor and a fixed quantization step size.

## 4.2 Recommendation performance

We evaluate the recommendation performance of the proposed meta compression algorithm by considering whether any of the top-$k$ recommendations for a given constraint performs at most within a factor of $\epsilon$ with respect to the optimal recommendation. We primarily refer to the task of maximizing accuracy subject to compression constraints, for which the analytical results in Section 3.2 hold. We also present some results for the task of maximizing compression given an accuracy constraint for models pre-trained in the next subsection.

### 4.2.1 Models pre-trained on CIFAR10

We compute top-$k$ recommendation accuracy by averaging it across several constraints. We also compute top-1 error, which characterizes how far off is the top recommendation from the optimal one in terms of compressed model accuracy (on average). For comparison purposes, we also consider selecting the single best pruning and quantization method on a dataset of problems and recommending the same for all architectures and compression level constraints (i.e., the static recommendation with ERM). We find that Slim pruning (Liu et al., 2017) and learned step size quantization (Esser et al., 2020) perform the best in such a setting.

We start by splitting the set of architectures into train set and test set. The meta prediction model is trained for architectures in the train split, and used to predict performance of architectures in the test split. We perform ten such splits to remove bias towards a specific choice. We set the tolerable drop in accuracy $\epsilon$ to 0.01 and vary the minimum compression constraint between 1x to 120x, with a step size of 10. We then

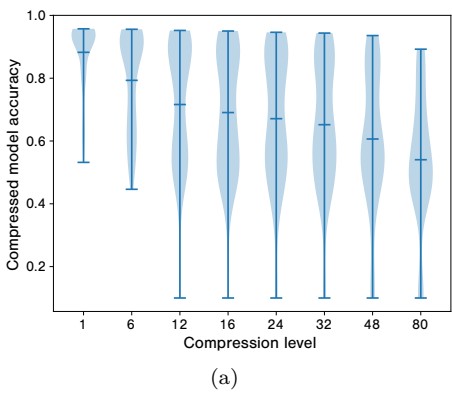 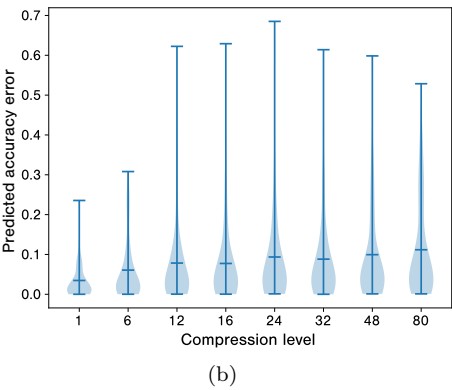

(a)                    (b)

Figure 4: Accuracy of the (a) compressed model and of the (b) prediction for different compression levels.

compare the predictions made using our meta prediction model against the optimal choice found through exhaustive search for each constraint. The corresponding results are reported in Figure 3a, which clearly shows how the proposed meta compression approach outperforms fixed recommendation with ERM.

Figure 3b reports different aggregate metrics across all compression constraints. The results show that the top-5 performance is 92% and the average error is only 0.01 compared to 0.25 for static recommendation with ERM. In other words, one the top-5 recommendations of meta compression is close to the optimal compression algorithm 92% of times and the performance of the recommended compression method has only a 1% difference in accuracy on average.

We also report the Kendall's Tau ($\tau$), which measures the rank correlation between the ordering induced by our predicted accuracy and the that of the observed accuracy across compression options under a given constraint (averaged over constraints). Higher values of $\tau$ indicate that the predictor preserves the relative orderings that drive recommendation quality: values close to 1 mean near-perfect agreement, while 0 implies no association. In our experiments, $\tau$ is high and comparable to the numbers reported by White et al. (2021) in a study on NAS. This is remarkable, as our study entails more complex settings that requires ranking compression methods applied to previously unseen architectures, compared to the simpler task of ranking unseen architectures in that work.

We also evaluate how the accuracy prediction model $g$ performs across different compression methods. We divide the test set by method under the same setup employed for Figure 3, then assess accuracy / prediction and recommendation performance on each subset. Table 1 reports a MAE ranging between 2 and 18%. Even though a few methods exhibit a rather high MAE, the top-5 recommendation remains high, indicating that $g$ preserves the within-method ranking of pruning / quantization levels even when absolute predictions are biased. Moreover, one-shot Prune + Quant has a higher MAE than most methods with retraining, possibly because retraining shifts and concentrates accuracy toward higher values, thereby making the prediction task easier. Overall, the recommendation quality remains strong across different methods.

Figure 4 characterizes how the compressed model accuracy and the prediction error vary with the compression levels (grouped into equally-sized bins). Figure 4a shows that accuracy are higher and more concentrated at very low compression levels, thereby reducing the prediction error. The mean prediction error remains roughly stable as compression increases, but its variance grows. Pushing compression beyond the range shown in the figure substantially reduces accuracy (below 50%), so raw accuracy values are not informative and are therefore omitted. Furthermore, Figure 4b shows the absolute error in the prediction accuracy across compression levels. The error remains consistently low, even at 48× and 80× compression. The variance is more pronounced in the higher range of the absolute error, although most of the values are concentrated below the mean. The high variance of the absolute error is notably correlated with that of the compressed model accuracy, as one would expect.

Finally, Figure 5 shows the accuracy in predicting the method that achieves the highest compression given an acceptable level of loss of accuracy. Specifically, Figure 5a show that the learned accuracy predictor

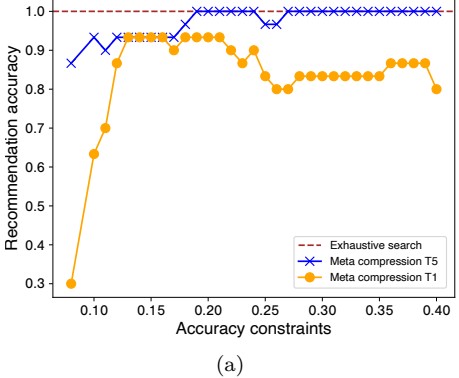

| | Meta Compression | |
|---|---|---|
| **Metric** | **Accuracy-constrained** | **Resource-constrained** |
| T5 Accuracy | 0.98 | 0.92 |
| T1 Accuracy | 0.84 | 0.66 |
| T1 Error | 0.10 | 0.01 |
| MAE | 0.07 | 0.10 |

(a)                                                    (b)

Figure 5: (a) Recommendation accuracy of meta compression against exhaustive search for different accuracy constraints and (b) summary metrics comparing recommendations for the accuracy-constrained compression maximization task with those for the compression-constrained accuracy maximization task.

performs similar to the previously-considered task, with the best top-5 recommendation performing close to the optimal solution across all constraint levels. Table 5b presents aggregate metrics, highlighting that meta compression achieves over 90% top-5 recommendation accuracy on both accuracy-constrained and resource-constrained tasks.

### 4.2.2 Models pre-trained on ImageNet

We conducted ImageNet recommendation experiments for a smaller set of compression levels due to significantly higher compute cost for fine tuning ImageNet models. The maximum sparsity level is set to 0.94, the considered quantization precision is limited to 16 and 32 bits, and the maximum compression level is set to 24. We have considered 10 different sparsity levels ranging from 0 to 0.94. We evaluate the recommendation performance of the proposed meta compression algorithm by considering whether any of the top-$k$ recommendations for a given constraint performs at most $\epsilon$ worse than the optimal recommendation. We find that the best static strategy for this case involves quantizing to 32 bits with LSQ and adapting Slim pruning rate to match the compression level provided in the constraint. We set $\epsilon$ to 0.01, same as the CIFAR10 setup, and vary the minimum compression constraint between 1x to 24x, with a step size of 1.

The top-1 recommendation performance and MAE of learned accuracy prediction function $g$ are reported in Table 2. The top-1 error reveals an 8% drop in accuracy on average, compared to the 44% drop when using the static ERM approach. The same observation can also be made from Figure 6, where meta compression significantly outperforms the static recommendation strategy at higher compression levels.

Table 3 shows the recommendation performance for different compression methods, while Figure 7 shows the distribution of compressed model accuracy and absolute error in accuracy prediction at several compression levels. Compared to the CIFAR10 setup, the accuracy of compressed models drops rapidly as compression level increases. This can be improved with more fine-tuning using more data and retraining epochs, at the cost of additional computation. Once again, the variance of compressed model accuracy increases with the mean absolute error in accuracy prediction at different compression level ranges. The absolute error values are concentrated below the mean, and the mean always stays below 0.12.

| Metric | Meta Compression | Static ERM |
|---|---|---|
| T1 Accuracy | **0.77** | 0.25 |
| T1 Error | **0.08** | 0.44 |
| MAE | **0.12** | 0.13 |

Table 2: Recommendation performance for the ImageNet setup.

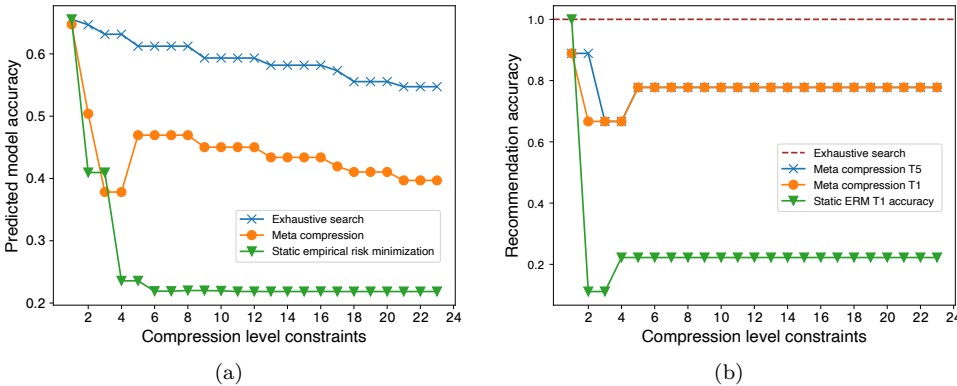

Figure 6: Accuracy of the (a) predicted model and of the (b) predictor across several compression level constraints.

| Compression Algorithms | T5 Accuracy | T1 Accuracy | T1 Error | MAE |
|---|---|---|---|---|
| Prune + Quant | 0.98 | 0.80 | 0.03 | 0.13 |
| Level-prune + Dorefa-quant | 0.69 | 0.49 | 0.16 | 0.25 |
| L1-prune + Dorefa-quant | 0.81 | 0.55 | 0.03 | 0.08 |
| TaylorFO-prune + Dorefa-quant | 0.73 | 0.66 | 0.02 | 0.08 |
| Slim-prune + Dorefa-quant | 0.90 | 0.79 | 0.04 | 0.19 |
| Level-prune + LSQ-quant | 1.0 | 1.0 | 0.01 | 0.10 |
| L1-prune + LSQ-quant | 0.95 | 0.79 | 0.02 | 0.06 |
| TaylorFO-prune + LSQ-quant | 0.97 | 0.75 | 0.02 | 0.06 |
| Slim-prune + LSQ-quant | 0.93 | 0.85 | 0.01 | 0.13 |

Table 3: MAE of the predictor $g$ trained for different compression algorithms for the ImageNet setup.

## 4.3 Design choice evaluation

This section evaluates the impact of data selection on performance and characterizes the generalization capability of the predictor.

### 4.3.1 Evaluation data

The primary metric under consideration when choosing the evaluation data $E$ is the generalization performance to test data. For satisfactory outcomes, it is desirable that the evaluation data contains samples not seen during the pre-training or fine-tuning phase. In the *ideal* scenario, this could be achieved by reserving some of the training data to be only used as evaluation data. However, this is often impossible as the full training data is generally used by pre-trained models, and pre-training the models again on a limited subset of train would be very expensive. Another option is to sample the evaluation data from the test data to avoid retraining from scratch. Unfortunately, this approach has several disadvantages. First, it risks leaking information from test data, possibly compromising the integrity of the final test evaluation. Second, sampling from test data in practice results in a dataset with a very limited size. Instead, we propose using diffusion models to generate evaluation data from the source dataset to overcome the above-stated limitations.

Figure 8a shows the top-5 recommendation accuracy as a function of the compression level. As expected, using evaluation samples drawn from the training set yields the worst performance across all compression levels. Sampling from the test set performs better but is typically inferior to the remaining two strategies. The ideal scenario (i.e., restricting evaluation to held-out samples from the original dataset) achieves strong results and sometimes outperforms using samples from the generative model at certain compression levels. However, evaluation data generated by a diffusion model trained on the training data delivers the best performance. Table 8b summarizes aggregate metrics for recommendation quality and MAE of the accuracy predictor, where using a diffusion model performs slightly better than the ideal scenario across all metrics.

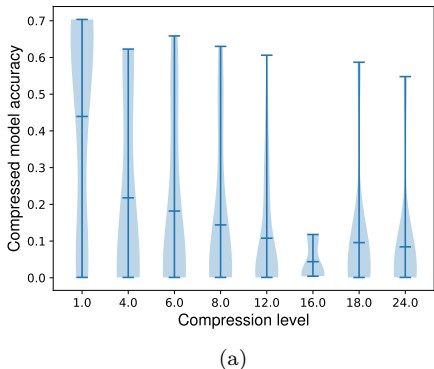 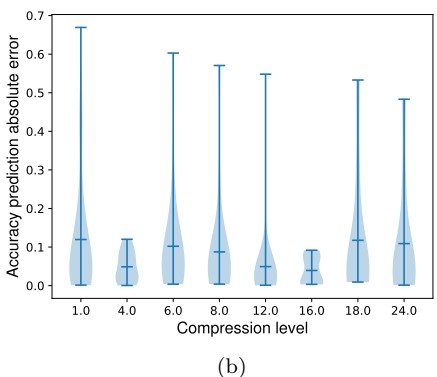

|   (a)   |   (b)   |

Figure 7: (a) Compressed model accuracy and (b) absolute error in accuracy prediction for different compression levels for the ImageNet setup.

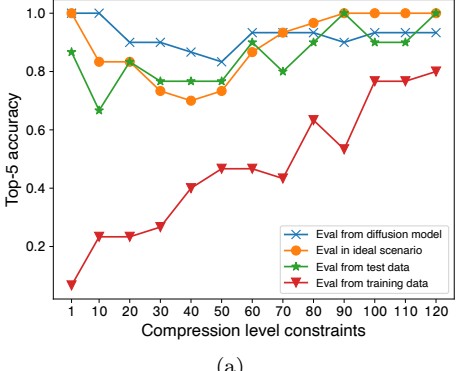

| Eval data selection | T5 Accuracy | T1 Accuracy | T1 Error | MAE |
|---|---|---|---|---|
| From training data | 0.48 | 0.14 | 0.13 | 0.24 |
| From test data | 0.85 | 0.55 | 0.06 | 0.12 |
| Ideal (From original dataset) | 0.89 | 0.58 | 0.03 | 0.12 |
| **Diffusion model samples** | **0.92** | **0.66** | **0.01** | **0.10** |

(b)

| Eval data selection | T5 Accuracy | T1 Accuracy | T1 Error | MAE |
|---|---|---|---|---|
| From training data | 0.35 | 0.14 | 0.30 | 0.17 |
| From test data | 0.78 | 0.77 | 0.08 | 0.12 |

(a) (c)

Figure 8: (a) Evaluation data selection choices. Prediction performance for different data selection strategies for models pre-trained on the (b) CIFAR10 and (c) ImageNet datasets.

We attribute this improvement to the ability to generate a sufficiently large evaluation set without reducing the size of the training set, which would otherwise degrade the performance of the pre-trained models.

For the ImageNet setup, we conducted experiments with two data selection choices, a) $E$ sampled from train data, and b) $E$ sampled from test data. Table 8c reports the obtained results. The behavior is consistent with the observation made for the CIFAR setup, that sampling from test data (and using a reduced test set for final evaluation) performs significantly better than sampling from train data as the feature evaluations done during the meta training phase generalize well to testing / recommendation phase.

### 4.3.2 Generalization performance

We train the meta predictor across multiple problems and compression algorithms. Therefore, it is very important to characterize how well it generalizes to scenarios involving unseen architectures, unseen data, or even novel compression algorithms. Such an analysis also reveals insights into what can be learned and what cannot, and is key to validating the hypothesis that performance of compression methods applied to novel problems can be predicted in practice (as stated by Theorem 3.2). We carried out several experiments to answer these questions, summarized in Table 4.

**New architectures.** This setting evaluates how well a meta predictor trained on a set of architectures selects compression methods for unseen architectures, assuming the evaluation data and method set match those used during meta training. On CIFAR10, splitting 12 pre-trained DNNs in a 3:1 train–test ratio and averaging across ten random splits yields only a 1% T1 error relative to the optimal method and a 92%

| New | | | CIFAR10 | | | | ImageNet | | | |
|---|---|---|---|---|---|---|---|---|---|---|
| Data | Archs | Methods | T5 Acc. | T1 Acc. | T1 Err. | MAE | T5 Acc. | T1 Acc. | T1 Err. | MAE |
| No | Yes | No | 0.92 | 0.66 | 0.01 | 0.10 | 0.78 | 0.77 | 0.08 | 0.12 |
| Yes | Yes | No | 0.91 | 0.66 | 0.02 | 0.11 | 0.78 | 0.77 | 0.09 | 0.13 |
| No | No | Yes | 0.86 | 0.34 | 0.11 | 0.13 | 0.74 | 0.51 | 0.14 | 0.14 |
| Yes | No | Yes | 0.85 | 0.34 | 0.12 | 0.15 | 0.74 | 0.51 | 0.15 | 0.14 |

Table 4: Generalization performance of $g$ to new data, architectures, and compression methods.

top-5 recommendation accuracy. On ImageNet, using a 2:1 split over 6 models and averaging across multiple random splits results in an 8% top-1 error. Despite fewer models, fewer compression levels, and using $E$ sampled from test data rather than a diffusion model, this is only slightly worse than the 6% top-1 error observed for CIFAR10 under the same conditions. These findings indicate that compression performance on novel architectures can be effectively predicted from sufficiently diverse architecture data.

**New data.** This setting tests robustness when the evaluation dataset used to compute meta features during training is unavailable or differs at test time. For CIFAR10, holding out the original evaluation set and using a disjoint dataset leads to an average 2% T1 error relative to the optimal method, with 91% top-5 accuracy, indicating appropriate generalization to dataset shift. For ImageNet, replacing the evaluation set similarly induces only a slight performance drop, mirroring the trend on CIFAR10 and suggesting robust generalization to new data across both benchmarks.

**New compression methods.** Here we assess generalization to unseen compression methods while recommending for pre-trained models already seen during meta training. We partition compression specifications by method, train on a subset, and evaluate on held-out methods while keeping the classifier set fixed. Across CIFAR10, top-1 selection performance degrades noticeably (higher T1 error or lower top-1 accuracy), yet top-5 accuracy decreases only modestly, indicating that while different methods have distinct trade-offs, shared structure often places the correct method within the top five. ImageNet exhibits the same pattern (see Figure 8a): a considerable top-1 degradation relative to the new-architectures setting but only a marginal reduction in top-5 accuracy. Overall, while unseen methods pose challenges for precise top-1 selection, the meta predictor maintains strong top-5 recommendation quality across datasets, reflecting shared regularities in compression-accuracy trade-offs.

## 4.4 Compute cost

This section characterizes the compute cost incurred by meta compression by breaking it down into the two main steps it entails, namely, meta training and meta recommendation. The section ends with an analysis of total cost, alongside a comparison with a compression-oriented NAS baseline.

### 4.4.1 Meta training cost

The dominant term in the cost of training the accuracy predictor $g$ is measuring the true accuracy $A$ of compressed models, because the fine-tuning required after pruning / quantization far outweighs both meta-feature extraction and updating $g$. The total cost therefore scales with the average cost of compressing one model times the number of meta training samples, i.e., (train-set architectures) × (compression settings $K_i$ per architecture).

On CIFAR10, fine-tuning a compressed classifier for one pruning / quantization setting (40 epochs) using 9 different compression algorithms takes about 40 minutes on a Tesla A100 GPU. Evaluating 10 sparsity and 4 quantization levels – 40 configurations in total – takes approximately 26 hours per architecture. Running 12 architectures in parallel on 12 A100s finishes in about 1.1 days of wall time (roughly 13 GPU days). On ImageNet, one pruning / quantization setting (40 epochs) takes about 300 minutes on an AMD MI250x. Evaluating 10 sparsity and 2 quantization levels – for a total of 20 configurations – approximately costs 100 hours per architecture. Training 6 architectures in parallel on 6 MI250x GPUs completes in about 4 days of wall time (roughly 25 GPU days).

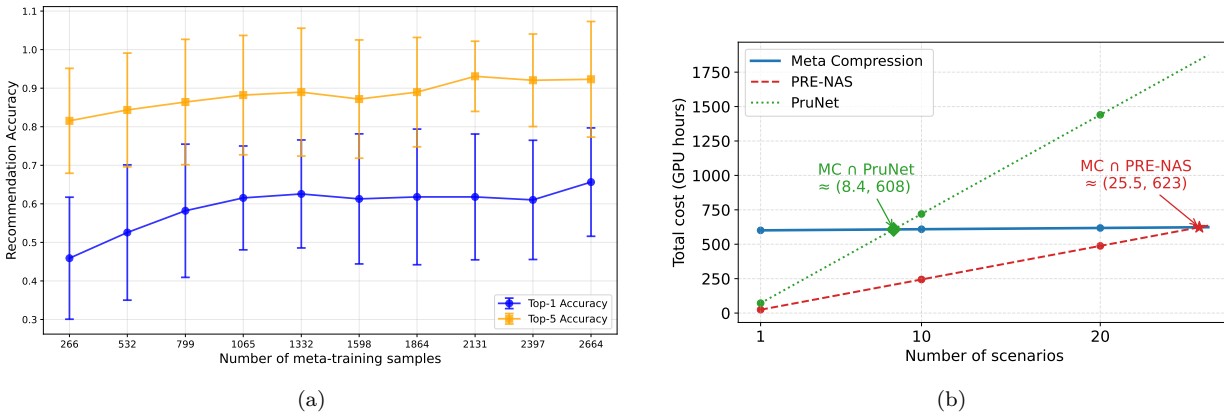

Figure 9: (a) Recommendation accuracy versus number of meta-training samples; (b) total cost comparison of meta compression with NAS-based alternatives.

### 4.4.2 Meta recommendation cost

Obtaining a meta recommendation for given pre-trained models and compression constraints requires extracting meta features (for instance, pre-trained accuracy) through a short inference pass – roughly 5 minutes for CIFAR10 and 20 minutes for ImageNet. Once features are available, the XGBoost predictor evaluates a compression method / level in about 20 milliseconds. By comparison, an exhaustive search over methods and levels would require several GPU days of fine-tuning compute.

### 4.4.3 Total cost

Since total cost scales with the number of meta-train samples, Figure 9a analyzes how sample count influences recommendation accuracy on CIFAR10. For each seed, architectures are split into train and test sets as in Section 4.2.1. We then incrementally add (architecture, compression setting) pairs to train the meta-prediction model and evaluate recommendation performance on the held-out test split. This process is repeated across 10 random seeds; the plot reports the average Top-1 and Top-5 accuracies along with their standard deviations (as error bars). Accuracy increases quickly until about 150 samples and then levels off; Top-5 accuracy exceeds 80% with 345 samples and reaches 92% with 3,456 (0.8 $\times$12 architecture$\times$9 compression methods$\times$40 compression levels) samples.

Figure 9b compares the total cost of delivering a trained architecture under scenario-specific resource constraints for meta compression and two NAS-based alternatives: PruNet (Kierat et al., 2022), a compression-oriented NAS method; and PRE-NAS (Peng et al., 2022), a general-purpose NAS method. Cost for NAS-based alternatives involves both the architecture search (as reported in the respective publications) and an estimate of the final training, set to 10 GPU hours for a fair comparison. The results show that NAS-based solutions are cheaper when considering a limited number of scenario-specific resource constraints, while meta compression becomes substantially more cost-effective as the number of deployment scenarios increases. Notably, the compression-oriented approach of PruNet is less effective than the generic search of PRE-NAS; this demonstrates that our approach is particularly suitable for explicit model compression.

## 5 Additional applications

This section examines the applicability of meta compression beyond image classification. We first quantify performance on a tabular regression task. We then discuss the feasibility of extending the framework to natural language processing, outlining how large transformer-based models should be handled.

| Metric | T1 Accuracy | T1 Error | MAE | Kendall Tau |
|---|---|---|---|---|
| Meta Compression | 1.00 | 0.00 | 0.13 | 0.51 |

Table 5: Meta recommendation and meta prediction performance for the California housing regression experiment.

### 5.1 Regression tasks

We evaluate applying meta compression to regression tasks to further demonstrate its applicability to diverse scenarios. For this purpose, we set up a smaller experiment with four benchmark architectures from (Gorishniy et al., 2021), namely, DCNv2, SNN, ResNet, and MLP. We employ the California housing regression dataset (Pace & Barry, 1997), a ratio of 3:1 between train and test data, 12 different compression levels, and consider the compressed regressor RMSE prediction instead of the compressed classifier accuracy prediction. Separate data splits were used for training, evaluation, and testing in the commonly used 64:16:20 ratio. RMSE values were normalized for convenience, as they are less than 0.51 for all pre-trained models.

Table 5 reports the obtained results. Meta compression achieves 100% top-1 recommendation accuracy when predicting the regression performance of a new architecture, clearly demonstrating the applicability of meta compression to the tabular datasets as well.

### 5.2 Natural language processing

As Figure 9 and the related discussion shows, training a meta predictor scales with the cost of compressing and evaluating architectures at target levels, including fine-tuning. In principle, no other constraints limit applying the meta compression framework to new architectures for scenarios other than image classification. For instance, meta compression could be as well applied to large language models (LLMs) in natural language processing. For state-of-the-art LLMs, however, training and fine-tuning costs are prohibitive, making them less suitable for direct application.

In fact, extending meta compression to LLMs presents unique challenges beyond computational cost. First, the scale disparity is substantial: while vision models in our study contain millions of parameters, frontier LLMs comprise tens to hundreds of billions, amplifying the cost of generating meta training samples by orders of magnitude. Second, LLM compression exhibits different characteristics than CNN compression – attention mechanisms, embedding layers, and feed-forward blocks respond heterogeneously to pruning and quantization, requiring richer meta features to capture these interactions. Third, performance evaluation is more complex: whereas image classifiers admit straightforward accuracy metrics, LLMs require assessment across perplexity, downstream task performance, and generation quality, complicating the definition of a unified prediction target.

Despite these challenges, several factors make LLM compression a compelling scenario for meta compression. The transformer architecture is remarkably homogeneous across model families, suggesting that meta knowledge learned from smaller models (e.g., 1–7B parameters) may transfer to larger variants within the same family. Moreover, the proliferation of on-device language assistants and privacy-preserving edge deployments creates precisely the diverse constraint landscape where meta compression excels – recommending optimal compression configurations across heterogeneous hardware without repeated evaluation. Recent advances in efficient LLM compression methods such as GPTQ (Frantar et al., 2022) and SparseGPT (Frantar & Alistarh, 2023) could serve as compression primitives within our framework. A practical approach would be to target smaller open-source LLM families, keeping meta-training costs tractable while still covering architectures representative of the broader LLM ecosystem.

## 6 Conclusion

This work has addressed the problem of compressing large deep neural networks (DNNs) so that they can be deployed onto edge servers and resource-constrained devices. Specifically, we learned how to compress DNNs with a novel meta learning approach – the first, to the best of our knowledge. Accordingly, we train

a regression model to predict the accuracy of a pre-trained DNN after being compressed with a combination of techniques – including pruning and quantization – without having to evaluate it. We take a flexible yet rigorous approach to achieve provably approximate correct meta learning. We also carry out an extensive evaluation using common compression schemes. The obtained results demonstrate that meta compression is effective, and that diffusion models are instrumental to improve the generalization capabilities and also the accuracy of the predictor. As a consequence, meta compression can be applied even to scenarios in which evaluation data is scarce, thereby extending its applicability in practice.

Our study has focused on classification tasks in the visual domain. Extending meta compression to different scenarios (such as speech recognition) and more complex architectures (including graph neural networks) would be an interesting direction for future work.

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

# A Meta Prediction Features

This section analyzes the impact of different features on the predictor accuracy by considering both the CIFAR10 and the ImageNet datasets.

The learned accuracy prediction model $g$ can offers valuable insights about the specific meta features that contribute the most towards accurate compressed model accuracy predictions. Specifically, decision trees allow us to compute feature importance by using the Breiman equation (Hastie et al., 2009), which are presented later in this section. First of all, we provide additional details about the configuration of the decision tree model and of the considered features.

## A.1 XGBoost meta prediction

The XGBoost model is configured with 100 estimators (i.e., the number of trees) and a maximum depth of 10 for each tree. The features are preprocessed as follows before being fed to the XGBoost model.

**Categorical Feature Encoding**. The raw data needs to be converted into a format that can be effectively used by the XGBoost model. To achieve this, we encoded the categorical features – including dataset identifier, compression method identifier – through one-hot encoding.

**Numerical Features**. Numerical features including loss and accuracy of the pre-trained classifier, gradient norms were incorporated after scaling them between 0 and 1. The number of architecture parameters was divided into 10 bins and linearly mapped onto the [0,1] interval.

## A.2 Feature importance

Figure 10 illustrates the feature importance. The results show that the prevalent features include the tags of a few compression algorithms, followed by the loss and accuracy of the pre-trained classifier. The target sparsity level is a stronger predictor than target quantization level, which is related to using more sparsity levels and fewer quantization levels in the considered configurations. Interestingly, the aggregate gradient metrics such as the $L_0$ and $L_1$ norm of gradients are more important than the number of parameters in the pre-trained model. This supports the observation that the slope of the solution learned after pre-training offers meaningful insights about the compression performance of the model. We also experimented with using more descriptive architecture features as well as the largest eigenvalues of Hessian to extract second-order derivative information; however, we observed no significant improvement in the results.

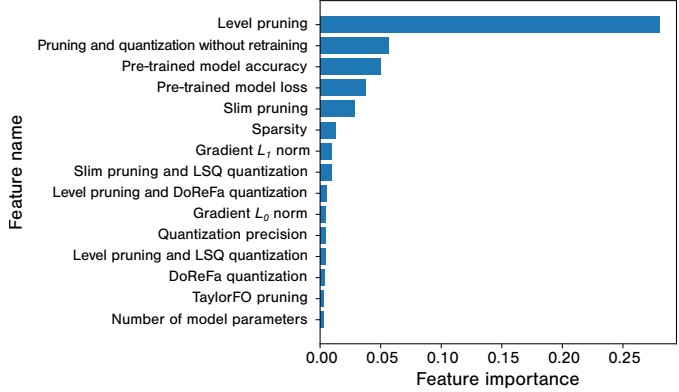

Figure 10: Feature importance for the meta predictor in the combined CIFAR10 and ImageNet setup.

