# OpenReview forum: "Meta Compression: Learning to Compress Pre-trained Deep Neural Networks"
_TMLR — Rejected by TMLR_

### Review · Reviewer_uDAA · 2025-10-12

**Summary Of Contributions:**

This work address deals with nerual network compression. Praticularly, the solve the challange in several compression algorithm where the perfomance can only measured posteriori. The authors achieve this by
1. The compression routine has 2 stages Pruning followed by the Quantization.
2. The approach uses diffusion model to obtain the Validation Set.
3.  The work trains neural networks called -Meta feature extraction and Accuracy Prediction model over range of purning sparsity and quantization levels.
4. These trained models are then used to predict the best compression method along with the prunig sparsity and quantization levels to fit the resource and accuracy constraints for a given problem.

The paper is generally well written and is easy to read and I believe the results  are promising.

**Audience:**

Yes

**Audience Explanation:**

The paper deals with Neural Network Compression and should be relevant to the audience.

**Broader Impact Concerns:**

I do not see any concern.

**Claims And Evidence:**

No

**Claims Explanation:**

1. The paper is motivated by stating current approaches have several hyperparameter (and requires hyper-parameter tuning ) or use NAS. Both would require significant compute. The proposed approach also requires significant compute to train meta learner (See Requested changes). Adding cost comparsion against optimal solutions for some of the methods should help this claim.
2.  The main paper only has results for a small scale dataset (CIFAR10) and all the evaluation is primarily on Convolution Networks. The authors claim this approach should be applicable to different domains/datasets/network achitectuers.

**Requested Changes:**

1. Unclear Experimental Setup - I do not understand how the meta feature extraction and accuracy perdictions modules are trained. Specifically,

    i. what are the inputs to the meta extractor. I understand it has architecture features and compression method features. Can the authors provide what are these features. (Like do architecture feature include number of layer, number of parameters, types of layers, and ways to encode the network connectivity ?). I believe this might be similar to (White et al. 2021), however, adding a few lines would help as the core approach of the paper is the accuracy estimation mechanism.

    ii.  I am unsure how authors train their network Section 4.1.1 mentions it is trained across 10 sparsity levels, 4 quatization levels and 12 architecutures. Do they train the meta extractor and accuracy predictors across several compression algorithms?  or Do we requires a separate network for each algorithm? As the predictions are not as accurate for testing with new algorithm, I would expect we will have to train across all the algorithms.
2. Comparsion - I believe authors use static ERM metioned section as authors mentions the numbers being better than this in several places. I think authors should evaluate there approach of compression with current state of the art, including different approaches for a give sparsity and accuracy constrainsts.  For examples, authors can use a state of the art compression approach without NAS  and run hyperparameter tuning; use a NAS based baseline. A comparsion of accuracies for these approaches will help understand the effectiveness of the approach.
3. Cost Comparsion Metric - Authors should consider adding the comparsion for training effort required for above comparsions. The proposed methods is compute intensive as it requires training over wide range of comperssion setting to train the meta networks. This seems similar to the effort required for hyperparamter tuning.
4. Datasets - Authors should consider including ImageNet results in the main paper. I believe showing results on larger more practical datasets will strengthen the paper.
5. Models - I believe authors can extend the study to Transformer based architectures like ViT. This will test the claim that the proposed approach can generalize to unseen architecture.
6. Generalizablility for new methods - Authors claim generalisability to new compressions methods is not important or practical use case for their approach. I do not understand  the rational behind this. If a new method provides superior performance and compression ratios (on say large datasets) over the supported methods (or the methods with which meta compression is trained with), generally new method would be preferred.
7. Tasks - Authors can also consider adding results on other domains like text classification - This should strengthen the claim of applicability to different domains.

---

> ### Author Response · Authors · 2025-12-09
>
> We thank the reviewer for the encouraging feedback and detailed comments. We address them next.
>
> _"Unclear experimental setup [...] what are the inputs to meta extractor [...] how authors train their network"_
>
> Appendix A.1 details the meta features employed for accuracy prediction and their importance. We also evaluated more expressive architecture features that encode network connectivity; they offered no meaningful performance gains, so we omitted them from the final model. New Section 4.4 in the revised article explains the settings employed for training the meta predictor.
>
> _"I am unsure how authors train their network [...] Do they train the meta extractor and accuracy predictors across several compression algorithms?"_
>
> Yes, the accuracy predictor is trained across several compression algorithms, with the compression algorithm identifier provided as a one-hot encoded vector for making the prediction.
>
> _"Comparison [...] with current state of the art [...] [such as using] a NAS based baseline"_
>
> We thank the reviewer for the suggestion. Accordingly, we have now added new results to compare meta compression against NAS-based alternatives, namely, PruNet (a compression-oriented NAS method) and PRE-NAS (a general-purpose NAS method).
>
> _"Cost comparison metric [...] for above comparisons"_
>
> We evaluated the total training cost of meta compression and the two NAS-based alternatives described above in Figure 9b of the revised paper. The figure shows that NAS-based solutions are cheaper when considering a limited number of scenario-specific resource constraints, while meta compression becomes substantially more cost-effective as the number of deployment scenarios increases. Notably, the compression-oriented approach of PruNet is less effective than the generic search of PRE-NAS; this demonstrates that our approach is particularly suitable for explicit model compression.
>
> _"Datasets: Authors should consider including ImageNet results in the main paper”_
>
> We agree with the reviewer that the paper would have been stronger if we presented the ImageNet results in the main text. Accordingly, we have merged them into Section 4 of the revised submission.
>
> _"Models: I believe authors can extend the study to Transformer based architectures like ViT"_
>
> Meta compression can indeed be extended to transformer-based architectures. The transformer architecture is remarkably homogeneous across model families, suggesting that meta knowledge learned from smaller models (e.g., 1-7B parameters) may transfer to larger variants within the same family. We have added these considerations to new Section 5.2 of the revised article.
>
> _"Generalizability for new methods: Authors claim [such] is not important or practical"_
>
> We understand the reviewer's concern and believe it arises from a misunderstanding. The analysis is Section 4.3.2 evaluates how well a trained meta predictor performs at predicting the performance of unseen compression methods. Different compression methods exhibit different compression-accuracy tradeoffs, making it difficult to predict the performance of unseen compression methods, as our results show as well. However, this does not mean that meta compression cannot be applied to novel compression methods. Incorporating novel compression techniques into meta compression requires training the meta predictor on data from the novel compression method to achieve a higher level of prediction accuracy. We have clarified the discussion in Section 4.3.2 accordingly to avoid any ambiguity.
>
> _"Tasks: Authors can also consider adding results on other domains"_
>
> Appendix C of the original submission reported results of applying meta compression to a regression task, in which models pre-trained on the California Housing dataset were considered. We have now moved the related discussion to Section 5.1 of the revised article and stressed the broader applicability of meta compression at the beginning of Section 3.1. We have also explained how to apply meta compression to natural language processing in the new Section 5.2.

---

> ### Comment · Reviewer_uDAA · 2025-12-13
> **Follow up: Have the authors uploaded the correct paper?**
>
> I believe authors might not have added the correct manuscript as per the response.
> I do not see the references in the rebuttal reflect in the main script.
> 1. Authors mention Appendix A.1  describes meta features. I located this as Appendix B.1
> 2. Authors mention "New Section 4.4 in the revised article explains the settings employed for training the meta predictor". I could not find section 4.4
> 3. authors mention adding results for NAS approaches, however I do not see them in the results Table 1
> 4. authors make reference to figure 9b. I could not find this.
> 5. Authors claim results for imagenet in moved to main paper. I still see it in appendix A
> 6. Cannot find section 5.2 which includes transformer results.
> 7. Section 5.1 showing broader applicability is missing.

---

> > ### Author Response · Authors · 2025-12-15
> >
> > The revised manuscript had already been uploaded when we posted our official comment and appears to be accessible via the PDF icon at the top of this page, as reflected in the revision history. It is possible there was a temporary issue; we would greatly appreciate it if you could kindly check again whether the revised manuscript is currently accessible on your end.

---

### Review · Reviewer_NWtM · 2025-10-21

**Summary Of Contributions:**

This paper investigates the problem of model compression from a meta-learning perspective. It argues that previous compression methods still require frequent retraining from scratch due to the vast search space. To address this issue, the paper proposes a meta-compression framework that directly predicts the performance of different compression strategies without the need for retraining, thereby significantly reducing computational costs. Experimental results on CIFAR-10 and ImageNet demonstrate the effectiveness of the proposed approach.

**Additional Comments:**

See above.

**Audience:**

Yes

**Audience Explanation:**

The topic of model compression is of significant importance to the TMLR community, and exploring this problem from a meta-learning perspective may offer a novel and valuable contribution.

**Broader Impact Concerns:**

N.A.

**Claims And Evidence:**

No

**Claims Explanation:**

The experimental results appear relatively weak and may not be sufficient to convincingly demonstrate the effectiveness of the proposed method.

**Requested Changes:**

1. The proposed meta-compression approach, which primarily aligns task losses, appears rather naïve, and its generalization capability remains doubtful.
2. The provided theoretical analysis can hardly be regarded as a convincing explanation of the method’s effectiveness. More detailed and specific analysis should be included to support the claimed contributions.
3. The experimental evaluation is relatively weak, as it is limited to CIFAR-10 and ImageNet, and lacks comparisons with state-of-the-art approaches.
4. Some typos exist, e.g., Unfortunately, these DNN so large that cannot fit into the limited resources of edge servers or end devices such as smartphones and IoT sensors.

---

> ### Author Response · Authors · 2025-12-09
>
> We thank the reviewer for recognizing the potential of our work and for the comments. We address them next.
>
> _"The proposed meta-compression approach [...] appears rather naïve, and its generalization capability remains doubtful"_
>
> We evaluate generalization across unseen architectures, datasets, and compression methods in Section 4.3.2. On new architectures, the meta-predictor yields 1% top-1 error and 92% top-5 accuracy. When the dataset changes, top-1 error is 2% with 91% top-5 accuracy, indicating limited degradation. These results are consistent with the analytical evaluation in Section 3.2, thereby substantiating the generalization capabilities of meta compression.
>
> _"The provided theoretical analysis can hardly be regarded as a convincing explanation of the method's effectiveness. More detailed and specific analysis should be included to support the claimed contributions"_
>
> We thank the reviewer for the opportunity to clarify. The purpose of Section 3.2 is to provide a theoretical foundation on the efficiency of our meta compression framework. In particular, Lemma 3.1 proves the scalability of meta learning, since the best compression method can be asymptotically obtained by considering only a logarithmic amount of training problems. Furthermore, Theorem 3.2 demonstrates that the compression recommendation function can be learned without any assumption on how the true data labels perfectly fit any hypothesis in the hypothesis class (i.e., agnostically). The theorem also proves that the best compression method can be found in polynomial time and establishes a bound on the maximum prediction error it can achieve. We have emphasized these contributions in the introduction and better explained the merit of the analytical characterization at the beginning of Section 3. We have also added new content (i.e., Figure 9a and the related discussion in Section 4.4.3) to empirically verify how meta compression scales as a function of the number of the meta-training samples.
>
> _"The experimental evaluation is relatively weak, as it is limited to CIFAR-10 and ImageNet, and lacks comparisons with state-of-the-art approaches."_
>
> Section 4.4 of the revised submission presents new results in terms of training cost in comparison with two NAS-based alternatives: PruNet, a compression-oriented NAS method; and PRE-NAS, a general-purpose NAS method. The corresponding results (Figure 9b) show that NAS-based solutions are cheaper when considering a limited number of scenario-specific resource constraints, while meta compression becomes substantially more cost-effective as the number of deployment scenarios increases. Notably, the compression-oriented approach of PruNet is less effective than the generic search of PRE-NAS; this demonstrates that our approach is particularly suitable for explicit model compression.
>
> _"Some typos exist, e.g., [...]"_
>
> We appreciate the detailed comment. We have fixed the typos and thoroughly proofread the text in the revised submission.

---

### Review · Reviewer_tT9h · 2025-11-24

**Summary Of Contributions:**

This paper addresses a practical bottleneck in deploying deep neural networks (DNNs) on edge devices. The authors propose “Meta Compression,” a framework for predicting the performance of a specific compression method without actually compressing the model and evaluating it. To achieve this, the framework extracts features from a pre-trained model and a candidate compression configuration (such as pruning sparsity or quantization bit-width) to estimate the final accuracy. They show that their method achieves 92% top-5 recommendation accuracy on CIFAR-10 and ImageNet.

Strengths:
- Once trained, the meta-predictor eliminates the need for expensive “trial-and-error” cycles to find the optimal compression strategy
- By using diffusion models to generate the evaluation data (E), they avoid data leakage issues
- The paper provides formal proofs regarding the error bounds of the recommendation strategy
- The predictor seems to generalize well to new architectures and datasets, not seen during the meta-training phase

Weaknesses:
- Generating the ground truth dataset to train the meta-predictor is expensive, requiring approximately 13 GPU days to run the necessary compression and fine-tuning loops for the architectures
- While the system handles new architectures well, the T1 accuracy falls significantly when the predictor encounters new compression methods
- The evaluation focuses primarily on image classification (CIFAR-10 and ImageNet), and the authors acknowledge that applying this to other domains like Natural Language Processing (NLP) or graph neural networks is left for future work

**Audience:**

Yes

**Audience Explanation:**

Yes. The paper is likely to be of interest to the TMLR audience, especially those interested in Edge AI, Efficient ML, AutoML and Neural Architecture Search (NAS) research. In addition, the paper shows an interesting application of diffusion models for creating “proxy” evaluation datasets to measure model robustness.

**Claims And Evidence:**

Yes

**Claims Explanation:**

Yes, the claims made are supported by accurate, convincing and clear evidence.

- The claim that meta compression accurately recommends compressions methods without running them is supported by the quantitative results comparing their predictions against and “Exhaustive Search” oracle
- The claim that diffusion models improve generalization and solve data scarcity is supported by the ablation study comparing four data selection strategies. They show that using diffusion samples yields the highest top-5 accuracy compared to other strategies, while preventing overfitting and preserving the integrity of test set.
- They also test architectures and datasets not seen during training to support their claim that the framework generalizes to new architectures and datasets
- By providing formal proofs they theoretically ground their approach

**Requested Changes:**

Major -

1. The authors claim “Meta Compression” as being a computationally efficient alternative to trial-and-error or NAS. However, it comes with a massive upfront cost. For practitioners with only a few models, exhaustive search would be cheaper than training the meta-predictor. Therefore, it would be good to include a “break-even” analysis that shows when this method becomes viable compared to other approaches. Perhaps a plot of “Number of Models to compress” vs “Total GPU hours” could be helpful
2. How does the meta-predictor handle a constraint that falls in between the discrete levels it was trained on? Does the system pick the next-closest configuration, or can it interpolate?
3. The current work is limited to vision (CNNs). Given the importance of LLMs and Transformers in the current landscape, adding a short discussion specifically on addressing the challenges of porting this to NLP (high cost of finetuning LLMs, etc.) would make the paper much more relevant to the community, in my opinion

Minor -
1. The link to the code in section 4 (first paragraph) is broken
2. Grammatical error in the abstract, “Unfortunately, these DNN so large that cannot fit into the limited resources of edge servers or end devices…”. Should be “Unfortunately, these DNNs are so large that they cannot fit into the limited resources…”

---

> ### Author Response · Authors · 2025-12-09
>
> We thank the reviewer for the supportive comments. We address them next.
>
> _"it would be good to include a ‘break-even’ analysis that shows when this method becomes viable compared to other approaches."_
>
>
> We evaluated the total compute (i.e., GPU-hours) as a function of the evaluated compression configurations and compared it against two NAS-based alternatives: PruNet, a compression-oriented NAS method; and PRE-NAS, a general-purpose NAS method. The corresponding results in Figure 9b of the revised paper show that NAS-based solutions are cheaper when considering a limited number of scenario-specific resource constraints, while meta compression becomes substantially more cost-effective as the number of deployment scenarios increases. Notably, the compression-oriented approach of PruNet is less effective than the generic search of PRE-NAS; this demonstrates that our approach is particularly suitable for explicit model compression. Section 4.4.3 of the revised submission presents these new results.
>
> _"How does the meta-predictor handle a constraint that falls in between the discrete levels it was trained on? Does the system pick the next-closest configuration, or can it interpolate?"_
>
>
> The framework interpolates beyond the discrete compression levels used in training. For any target compression constraint, we first enumerate feasible pruning–quantization combinations that satisfy it. The trained meta-predictor – a regression model – then estimates accuracy for arbitrary pruning and quantization levels. We recommend the configurations with the highest predicted accuracy for actual compression. We have clarified this aspect at the end of Section 3.3.
>
> _"adding a short discussion specifically on addressing the challenges of porting this to NLP (high cost of finetuning LLMs, etc.) would make the paper much more relevant to the community"_
>
> As Figure 9b shows, training a meta predictor scales with the cost of compressing and evaluating architectures at target levels, including fine-tuning. In principle, no other constraints limit applying the meta-compression framework to new architectures. For state-of-the-art LLMs, however, training and fine-tuning costs are prohibitive, making them less suitable targets. Nevertheless, recent advances in efficient LLM compression methods (such as GPTQ and SparseGPT) would make meta compression viable for targeting smaller open-source LLM families. We have added these considerations to new Section 5.2 of the revised article.
>
> _"The link to the code in section 4 (first paragraph) is broken"_
>
> Thank you for bringing this to our attention. The shared link had expired; it has now been reactivated and is accessible. Please let us know should you encounter any further issues in accessing the code.
>
> _"Grammatical error in the abstract [...]"_
>
> We appreciate the detailed feedback. We have resolved the reported issues and thoroughly proofread the revised article.

---

### Decision · Action_Editor_QBQ1 · 2026-01-01

**Recommendation:** Reject

**Additional Comments:**

Given the substantial improvement in the paper and the lack of consensus from the reviewers, I believe it is appropriate to allow a resubmission of a major revision if the authors so wish, and would encourage the authors to consider this if they believe they are able to address the final feedback presented here.

I also want to apologize again to the authors on the extended review timeline in this instance. Unfortunately with a reviewer dropping out late in the reviewing timeline despite having acknowledged the review, and the subsequent end of the year/holiday delays, this was  unavoidable, but generally TMLR seeks to provide reviews on a shorter timeline than was achieved here.

**Audience:**

Yes

**Audience Explanation:**

The reviewers were consistent in their evaluation that the paper would be relevant to some of the TMLR audience, I believe this is best summarized by Reviewer tT9h: "The paper is likely to be of interest to the TMLR audience, especially those interested in Edge AI, Efficient ML, AutoML and Neural Architecture Search (NAS) research. In addition, the paper shows an interesting application of diffusion models for creating “proxy” evaluation datasets to measure model robustness."

**Claims And Evidence:**

No

**Claims Explanation:**

In their initial reviews, the reviewers were consistent in their view that this criteria was not satisfied, but post-rebuttal the reviewers were split. In particular, Reviewer tT9h appreciated the substantial revisions made during rebuttal, pointing out that the meta-learning was better motivated with the inclusion of a computational cost break-even analysis, and discussion on LLMs, amongst other corrections addressed their concerns.

However, Reviewer uDAA and NWtM remained unconvinced by the empirical results post-rebuttal. Reviewer uDAA did not find the baselines to be appropriate, finding the consistency across compression levels troubling without an explanation, and in particular pointing out that iterative pruning is a strong baseline that should be considered. Reviewer uDAA also found the comparison between NAS and the proposed meta-compression method problematic in their different methodology, and the details of the NAS results presented. Reviewer NWtM while finding meta-compression generally well-motivated, was unconvinced by both the theoretical and empirical evidence presented, in particular the limited datasets the results were evaluated on, and did not find the rebuttal comments/changes to address these weaknesses.

I believe it's fair to say that the paper has seen substantial improvements over the review process, and is in a much better position at the end of the review process. Despite this, given that most reviewers found the empirical and theoretical evidence presented unconvincing post-rebuttal, and that it appears not all the requested feedback/changes were addressed on this front, I must conclude that the claims presented in the paper are not currently supported by convincing and clear evidence.

**Resubmission Of Major Revision:**

The authors may consider submitting a major revision at a later time.